# An Injectable Engineered Cartilage Gel Improves Intervertebral Disc Repair in a Rat Nucleotomy Model

**DOI:** 10.3390/ijms24043146

**Published:** 2023-02-05

**Authors:** Basanta Bhujel, Soon Shim Yang, Hwal Ran Kim, Sung Bum Kim, Byoung-Hyun Min, Byung Hyune Choi, Inbo Han

**Affiliations:** 1Department of Biomedical Science, College of Life Sciences, CHA University, Seongnam 13496, Republic of Korea; 2ATEMs Inc., Seoul 02447, Republic of Korea; 3Department of Neurosurgery, Kyung Hee University, Dongdaemun-gu, Seoul 02447, Republic of Korea; 4Wake Forest Institute of Regenerative Medicine, School of Medicine, Wake Forest University, Winston Salem, NC 27101, USA; 5Department of Biomedical Sciences, Inha University College of Medicine, Incheon 22212, Republic of Korea; 6Department of Neurosurgery, CHA Bundang Medical Center, School of Medicine, CHA University, Seongnam 13496, Republic of Korea

**Keywords:** cartilage gel, intervertebral disc degeneration, regeneration, human fetal cartilage-derived progenitor cells, extracellular matrix

## Abstract

Lower back pain is a major problem caused by intervertebral disc degeneration. A common surgical procedure is lumbar partial discectomy (excision of the herniated disc causing nerve root compression), which results in further disc degeneration, severe lower back pain, and disability after discectomy. Thus, the development of disc regenerative therapies for patients who require lumbar partial discectomy is crucial. Here, we investigated the effectiveness of an engineered cartilage gel utilizing human fetal cartilage-derived progenitor cells (hFCPCs) on intervertebral disc repair in a rat tail nucleotomy model. Eight-week-old female Sprague-Dawley rats were randomized into three groups to undergo intradiscal injection of (1) cartilage gel, (2) hFCPCs, or (3) decellularized extracellular matrix (ECM) (*n* = 10/each group). The treatment materials were injected immediately after nucleotomy of the coccygeal discs. The coccygeal discs were removed six weeks after implantation for radiologic and histological analysis. Implantation of the cartilage gel promoted degenerative disc repair compared to hFCPCs or hFCPC-derived ECM by increasing the cellularity and matrix integrity, promoting reconstruction of nucleus pulposus, restoring disc hydration, and downregulating inflammatory cytokines and pain. Our results demonstrate that cartilage gel has higher therapeutic potential than its cellular or ECM component alone, and support further translation to large animal models and human subjects.

## 1. Introduction

Chronic lower back pain (LBP) is a major complication acquired from intervertebral disc (IVD) degeneration, resulting in a significant burden on individuals and society. IVD degeneration is influenced by genetic, mechanical, and nutritional factors; thus, it is recognized as a complex and multifactorial process [1,2,3,4,5,6,7]. The depletion of proteoglycans in the nucleus pulposus (NP), reduced disc hydration, and disorganization of the lamellar collagen fiber network within the annulus fibrosus (AF) are the initial marks of degeneration [8,9]. Discectomy is the most frequently performed spinal surgery for lumbar discal herniation to relieve the pressure on a spinal nerve root by eliminating the herniated disc causing the leg pain. However, discectomy does not repair AF defects caused by NP herniation, and 10–30% of discectomy patients experience further disc degeneration, reherniation, and recurrent pain [10,11]. Furthermore, all IVD defects cause progressive IVD degeneration, with a high likelihood of disability due to a reduction in IVD height and biomechanical instability [5]. Therefore, there is a urgent clinical need to evolve AF repair strategies after discectomy to restore IVD function and avoid repeated pain. Mechanical AF closure devices, including Xclose™ (Anulex Technologies, Minnetonka, MN, USA) and Barricaid^®^ (Intrinsic Therapeutics, Woburn, MA, USA), have been flourished to inhibit reherniation after discectomy. However, these devices do not improve tissue healing and thus are inefficient to prevent further disc degeneration during long-term follow-up [10]. Therefore, these modalities are not totally competent to restore the structural and functional properties of the degenerated disc after discectomy [12].

A study involving only simple intradiscal injections of biomaterials demonstrated promising results for IVD regeneration by restoring the structure and function through spontaneous spheroid formation, without any transplantation of stem cells or administration of growth factors [13]. Delivering cells to the IVD cavity after discectomy using injectable biomaterials could be a more promising approach for IVD repair [13,14,15,16,17,18,19,20,21,22,23,24,25,26]. Several preclinical and clinical studies have demonstrated that the incorporation of cells with biomaterials into the IVD induces IVD regeneration by increasing NP hydration, increasing synthesis of the extracellular matrix (ECM), suppressing inflammation, and reducing pain [10,27]. Injectable biomaterials strengthen the cellular regeneration and proliferation ability in the injured IVDs. They also suppress cell leakage from the site of injection, as they provide instant biomechanical stabilization to injured discs [17]. To target both the NP and AF after discectomy, an injectable repair strategy utilizing two distinct biomaterials is considered a suitable approach to treat degenerated IVDs. In a study, a combined approach for NP augmentation using hyaluronic acid (HA) injection and a photo-crosslinked collagen patch for repairing AF after discectomy prevented IVD degeneration by healing defects in the AF, restored water content to the NP, and protected the native mechanical properties of IVDs [16].

We have previously developed an injectable cartilage gel, consisting of human fetal cartilage-derived progenitor cells (hFCPCs) and cartilage extracellular matrix (ECM) produced by hFCPCs [28,29]. The hFCPCs have high proliferation and colony-forming abilities and differentiate well into chondrocytes and other mesengenic lineages [30,31,32]. They produce a high quantity of ECM molecules, such as collagen and glycosaminoglycans (GAGs), and can be used for cartilage tissue repair and regeneration of other tissues [33]. It has also been confirmed that hFCPCs are immune-privileged and have immune-modulatory and anti-inflammatory effects [34]. However, mesenchymal stem cells (MSCs) have inadequate differentiation ability, limiting their potential to meet clinical needs for tissue regeneration; they demonstrate phenotypic drift during long-term maturation, conflicting their mass production. Similarly, chondrogenic differentiation of MSCs for cartilage regeneration has a potential risk for hypertrophy and ossification [35,36]. In opposition to this, stem or progenitor cells from fetal tissues may be a substitute for MSCs. They demonstrate a considerable proliferative and differentiation potentiality, compared to MSCs [28,31]. Chondrocytes are capable of surviving and differentiating within the harsh IVD environment, most probably due to their similarities to the condition of the cartilage where the chondrocytes are derived. They are vital for the synthesis of two major constituents in IVD, the matrix collagen and proteoglycan [37]. In human nasal chondrocytes (NC), the transcription factor forkhead box protein 1 (FoxF1) was detected, which is involved in cell growth and ECM regulation in anti-fibrotic pathways, but it was not detected in MSCs, thus establishing a genetic similarity to NP cells [38]. 

In a full-thickness cartilage defect model, we confirmed that the injectable engineered cartilage gel demonstrated good therapeutic potential for articular cartilage regeneration [28]. There were no adverse events, and administration of the cartilage gel led to an increase in collagen and GAG contents and exhibited potential for chondrogenic remodeling after implantation. The cartilage gel also bore clinical benefits in terms of its injectable and adhesive mechanical properties with the aggregate modulus of approximately 10 kPa and adhesion strength of 1.8 kPa [28]. The adhesive nature of the cartilage gel on irregularly shaped defects would lead to a fixation-free, seamless fit into the defect area. We thought that the advantages of the engineered injectable scaffold-free cartilage—namely, its gel-pliable properties, low immunogenicity, easy application and integration into the defects, and effective cartilage tissue regeneration could also be beneficial for disc regeneration. Our strategies for the clinical use of the cartilage gel are the prevention of further degeneration and restoration of IVD structure and function after discectomy in patients who require discectomy due to severe IVD herniation. Therefore, we hypothesized that the injectable cartilage gel could restore IVD structure and function in a nucleotomy model by altering matrix turnover, downregulating inflammation and pain, even in severely unfavorable surroundings for cell survival. In addition, we compared the benefits of implanting the cartilage gel with that of the cell source (hFCPCs) or decellularized ECM alone. 

## 2. Results

### 2.1. Cartilage Gel Exhibited Anti-Allodynic Effects

Immediately after making a nucleotomy model in Co4-5 and Co5-6, we injected the cartilage gel, hFCPCs, or ECM into the Co4-5 discs of each group, and we used the Co5-6 disc as a defect-only group (Figure 1A). To demonstrate the mechanical allodynia caused by nucleotomy, a von Frey filament was applied to the ventral base of the tail. The von Frey test was performed 2 days before (D-2) and 2, 7, 14, 21, 28, 35, and 42 days after surgery (Figure 1B). The increase in the pain sensitivity could result from the painful degenerative disc [39]. No significant differences between the three treatment groups were observed at the baseline (before surgery). The 50% withdrawal threshold (g) was significantly lower in the rats treated with the ECM or hFCPCs than in the rats treated with cartilage gel (one-way ANOVA, *** *p* < 0.001). However, the 50% withdrawal threshold (g) was significantly higher in the cartilage gel-treated group at 42 days, indicating that it significantly alleviated mechanical allodynia in the rat tail. 

### 2.2. Cartilage Gel Implantation Restored the Disc Anatomy and Hydration in the Rat Tail Nucleotomy Model 

In vivo, we performed T2-weighted magnetic resonance imaging (MRI) six weeks (42 days) after implantation, to study the disc anatomy and water content in IVDs in the rat coccygeal disc. The MRI index was calculated for the healthy discs (Co3-4), injury-only discs (Co5-6), and the discs treated with cartilage gel, hFCPCs, or ECM (Co4-5) (Figure 2A,B). On coronal and axial T2-weighted MRI, the MRI index was found to be higher in the cartilage gel-implanted discs. By contrast, the hFCPC- and ECM-treated discs were found to have lower MRI index values than the cartilage gel-treated discs. The Pfirrmann grade was significantly lower in cartilage gel-treated discs in comparison to the injury-only, ECM, or hFCPCs-treated discs (one-way ANOVA *** *p* < 0.001) (Figure 2C). Overall, the MRI findings suggested that cartilage gel treatment preserved the water content and enabled the best restoration of disc anatomy with enhanced disc regeneration.

### 2.3. Cartilage Gel Protected the Proteoglycan Content by Delaying the Loss of Cell Number and Positive Matrix Area in the IVD of the Rat Tail Nucleotomy Model 

After MRI analysis, we then assessed the harvested discs through histological analyses. We performed safranin-O and hematoxylin and eosin (H&E) staining to demonstrate proteoglycan distribution in the NP and overall IVD morphology (Figure 2D). We calculated histological scores along with the disc NP area and NP cell number. In the injury-only disc tissues, H&E staining demonstrated annular rupture and mixed clustering of NP cells, whereas the healthy discs had well-organized AF tissue, normal NP cellularity, and no cell clusters. The safranin-O and H&E staining intensities of the NP were significantly reduced, whereas the histological grade notably increased in the injury-only discs, indicating less preservation of disc morphology and proteoglycan content. However, this result was reversed in cartilage gel-treated discs, which presented a significantly increased NP area and cell number in H&E staining, as well as the lowest histological score, which indicated that cartilage gel was capable of preserving the disc structure and proteoglycan content in comparison with the hFCPCs or ECM-treated discs (Figure 2E,F,G). Meanwhile, the injury-only discs without any treatments demonstrated a gross loss, increased clefts, and progressive collapse of the NP matrix. Furthermore, we analyzed the cell number from H&E staining, and the results demonstrated a reduction in cell loss in the NP region of the cartilage gel-injected discs compared to the hFCPCs and ECM-treated discs. In addition to this, a delayed loss of the H&E-positive disc NP area was also observed in the cartilage gel-injected discs relative to the hFCPCs and ECM-treated discs (Figure 2G) (one-way ANOVA, *** *p* < 0.001). The observation of marked histomorphological differences between the cartilage gel and the other treated groups suggests the future potential of cartilage gel for viable disc tissue repair.

### 2.4. Cartilage Gel Preserves the Matrix Proteins in the Disc NP of the Rat Tail Nucleotomy Model

One of the major consequences of IVD degeneration is the massive destruction of ECM proteins, such as aggrecan and type II collagen, in the IVD. Proinflammatory cytokines, including tumor necrosis factor-alpha (TNF-α) and interleukin-1-beta (IL-1β), can markedly induce the production of matrix metalloproteinases (MMPs), leading to significant decreases in aggrecan and type II collagen [40]. Our experiments revealed significantly higher expression of aggrecan and type II collagen in the cartilage gel-treated discs than in the hFCPCs or ECM-treated discs. By contrast, the expression of aggrecan and type II collagen was dramatically diminished in the injury-only discs (one-way ANOVA, *** *p* < 0.001). However, the highest expression was observed in healthy discs (Figure 3(Ai,ii),B,C). Hence, these investigations revealed that the cartilage gel had the potential to restore matrix components in the degenerated disc and contributed to disc repair.

### 2.5. Cartilage Gel Preserved the Endogenous Disc NP Progenitor Cells in the Rat Tail Nucleotomy Model

Next, we performed immunofluorescence staining for brachyury and Tie 2 to assess cells with endogenous phenotypes. The results indicated the presence of cells with brachyury-positive or Tie 2-positive disc NP phenotypes, supporting the accelerated migration of remnant cells, encouraging the original disc NP phenotype maintenance and contribution to the repair process in nucleotomy-induced disc damage [13]. Further, brachyury positively regulates aggrecan in human and rat NPCs [13]. High expression levels of Tie2 and brachyury were observed in the healthy discs. Interestingly, a significant increment in the expression of brachyury and Tie 2 was observed in cartilage gel-treated discs in comparison to hFCPCs and ECM-treated discs. However, this expression was dramatically reduced in the injury-only discs (Figure 4A,B) (one-way ANOVA *** *p* < 0.001). Hence, these results demonstrated that cartilage gel potentially contributed to the repair process of nucleotomy-induced disc damage by inducing the expression of brachyury and Tie2 in the degenerated discs.

### 2.6. Evaluation of Cell Proliferation in Response to Cartilage Gel in the Disc NP of the Rat Tail Nucleotomy Model

Immunofluorescence staining for human nuclei antigen was performed to evaluate the proliferative effect of cartilage gel in IVD tissue. Our findings indicated that the cartilage gel was the most suitable candidate, demonstrating significantly higher expression of the human nuclei antigen than the hFCPCs and ECM-treated discs, indicating that implanted cells in the cartilage gels still survived six weeks post-implantation (one-way ANOVA, *** *p* < 0.001). In the previous study, engineered cartilage cells were observed and differentiated after in vivo transplantation and the presence of human origin cells filling the defect after 24 weeks was observed in non-human primate model investigation. However, a study utilizing human umbilical cord blood derived MSC hyaluronic acid gel composite in a rabbit model resulted in the disappearance of human origin cells by the 16th weeks [28]. We were unable to detect human nuclei in the healthy discs and injured discs (Figure 5A,B).

### 2.7. Cartilage Gel Downregulated Catabolic Enzymes in the Disc NP of the Rat Tail Nucleotomy Model 

Next, we performed immunofluorescence staining in rat tail discs to investigate the expression of MMP-13, which plays a vital role in the degradation of ECM proteins, including aggrecan and type II collagen, during the progression of IVD degeneration [41]. In a previous study, the expression of MMP-13 was found to be diminished in the control discs but upregulated in degenerated discs [42]. In the present study, the lowest expression of MMP-13 was observed in healthy discs, followed by those treated with the cartilage gel, compared to the hFCPCs and ECM-treated discs. On the contrary, its expression was significantly higher in the injury-only discs (Figure 6A,C) (one-way ANOVA *** *p* < 0.001). Hence, these results demonstrated that cartilage gel may be a potential candidate for downregulating catabolic enzymes in degenerated discs and may contribute to the disc repair in the degenerated IVD.

### 2.8. Cartilage Gel Downregulated Proinflammatory Cytokines in the Disc NP of the Rat Tail Nucleotomy Model

Likewise, immunofluorescence staining for TNF-α and Il-1β in the IVD was performed to investigate the effect of inflammatory signaling in our model. Productions of these cytokines enhance rapid degradation of ECM, production of chemokine, and alterations in the IVD phenotype. They have potent pro-inflammatory effects, stimulating the release of pro-inflammatory factors and interrupting the harmony between catabolic and anabolic metabolism in the IVD [43,44]. Previous studies have found that the expression of TNF-α and Il-1β in degenerated IVDs was higher than in non-degenerated IVDs [45,46,47]. Likewise, in our study, the lowest expression levels of TNF-α and Il-1β were observed in cartilage gel-treated discs in comparison to hFCPCs and ECM-treated discs. Conversely, this expression was significantly higher in injury-only discs and minimal in the healthy discs (Figure 6B,D) (one-way ANOVA *** *p* < 0.001). Hence, these results demonstrated that the cartilage gel has the potential to downregulate proinflammatory cytokines in degenerated discs and help in disc regeneration. 

### 2.9. Cartilage Gel Reduced Injury-Induced Pain in the Disc NP of the Rat Tail Nucleotomy Model 

Further, we performed immunofluorescence staining for sensory neuropeptide, calcitonin gene receptor protein (CGRP), which acts as a pain modulator. A prior study found that CGRP and its receptors were activated abnormally in degenerated IVDs in comparison to normal IVDs [48]. Additionally, CGRP is known to be involved in sensitization of nerves, with increased expression indicated after injury [49]. We demonstrated the expression of CGRP in healthy, injury-only, cartilage gel-treated, hFCPCs-treated, and ECM-implanted discs (Figure 7A). The immunoreactivity of CGRP in the healthy discs was lowest, followed by cartilage gel-treated discs. Likewise, the hFCPCs and ECM demonstrated significantly increased expression of CGRP, similarly to the injury-only discs. The expression of CGRP was significantly higher in the injury-only discs (one-way ANOVA, *** *p* < 0.001) (Figure 7A,B). Thus, these results confirmed that cartilage gel treatment reduced injury-induced pain in IVD. 

### 2.10. Cartilage Gel Reduced mRNA Levels of Pro-Inflammatory Cytokines in the Rat Tail Nucleotomy Model 

Next, we evaluated the mRNA expression of inducible nitric oxide synthase (iNOS) and TNF-α in the degenerated disc. The RT-qPCR results demonstrated significantly upregulated mRNA levels of TNF-α and iNOS in the injury-only discs. However, the expression was significantly reduced in discs treated with the cartilage gel as compared to the hFCPCs or ECM-treated discs (Figure 8A,B) (one-way ANOVA, ** *p* < 0.01). This indicates that the cartilage gel can prevent ECM degradation via reduced expression of pro-inflammatory cytokines, eventually helping in IVD regeneration.

## 3. Discussion

hFCPCs are considered an attractive candidate for the restoration or regeneration of damaged articular cartilages [28,29,30,31,34]. In our previous studies, we utilized the tissue-engineered cartilage gel consisting of hFCPCs and cartilage ECM to regenerate defective cartilage in the monkey knee [28]. The development of an injectable cell-laden gel is essential to improve IVD repair in translation to clinical use. With this aim, we have demonstrated the therapeutic potential of injectable engineered cartilage gel in a rat tail nucleotomy model. The primary findings of this study are as follows: the cartilage gel (1) exhibited anti-allodynic effects, (2) restored the disc anatomy and hydration of IVDs, (3) protected the proteoglycan content by delaying the loss of cell number and positive matrix area in the IVD, (4) preserved the matrix proteins in the disc NP, (5) downregulated the catabolic and proinflammatory cytokines in the disc NP, (6) preserved the endogenous disc NP progenitor cells, (7) inhibited injury-induced pain in the disc NP, and (8) reduced the mRNA levels of pro-inflammatory cytokines. Thus, these findings demonstrate the potential of the cartilage gel for the regeneration of degenerated IVDs. 

In the present study, we used the cartilage gel or hFCPC-seeded cartilage matrix. The cartilage gel demonstrates the characteristics of both mesenchymal stem cells (MSCs) and chondrocytes, having high proliferation, differentiation (chondrogenesis, adipogenesis, and osteogenesis), anti-inflammation activity, and cell migration abilities in vitro and in vivo; it indicates clinically useful properties, including high adhesiveness, plasticity, and continued chondrogenic remodeling after implantation in the defect site. It is often challenging to acquire sufficient levels of type II collagen in adult MSCs-based systems in comparison to cartilage gel [28,29,34]. In our previous study, in a non-human primate cartilage defect model, cell distribution of the cartilage gels after 24 weeks indicated transplanted cell remaining within the defect [28]. In a study by Quintin et al., the results demonstrated that human fetal femoral head cells were capable of differentiating well into chondrogenic, adipogenic, and osteogenic lineages [50]. A study reported that in combination with cartilage-derived stem cells with a molecular therapeutic (NTG-1010), when treated in a rat-tail model of degenerative disc disease, the results demonstrated robust cellularity, maintenance of disc height and size of the NP, decreased levels of matrix degrading enzymes (e.g., MMP-3 and MMP-13) and pro-inflammatory cycloxygenase-2, and that the levels of type II collagen and aggrecan are largely unregulated [51]. Our study also found that the cartilage gel-injected discs demonstrated an overall decrease in inflammation with an increase in matrix proteins. Likewise, in a study performed in New Zealand White rabbits of the IVD degeneration model, when autologous chondrocytes from auricular cartilage were implanted into the IVD in the place of removed nucleus pulposus, the result demonstrated a survival of a significant percentage of implanted chondrocytes and produced specific hyaline-like cartilage. Further, implanted autologous chondrocytes not only survived for six months following implantation, but also maintained the metabolic phenotype [52]. In an in vitro study, human fetal cells isolated from fetal cartilaginous tissues demonstrated high chondrogenic differentiation, increased aggrecan and type II collagen production, and decreased cycloxygenase-2 expression. Fetal cells were found to be positive for the surface markers CD44, CD73, CD105, and CD166 and were negative for CD34 and CD45, like MSCs. Similarly, the potential of fetal cells to survive in low-oxygen and low-glucose conditions, their low plasticity, and their anti-inflammatory properties are interesting assets for IVD regeneration [53]. Similarly, in research using human fetal chondrocytes in an in vivo model, no evidence of immunologic rejection was detected, and they demonstrated a more favorable safety profile than stem cells [54]. Nasal septal cartilage has also been found to be an alternative source of therapeutic cells for IVD regeneration in terms of high cell viability and ECM accumulation in the presence of a growth factor, transforming growth factor (TGF)-β3 [55]. However, in our study, we were able to restore the IVD cavity after removal of the NP without any exogenous growth factor supplementation using the cartilage gel. Furthermore, cells from juvenile articular cartilage appeared to be the best cell source compared to undifferentiated cells in the porcine lumbar IVD degeneration model. They were capable of producing adequate cartilage-like matrix, high levels of the anti-angiogenic/neurogenic factor, chondromodulin-I, surviving at least 12 months post injection within the IVD environment with no apparent adverse effects, thereby suggesting that they could be a better cell source than native MSCs [37]. It was observed that autologous chondrocytes from articular cartilage survived and produced hyaline-like cartilage six months after transplantation into injured rabbit discs. The promising chondrogenic potential of the foetal cells from human foetal spine tissue was observed in vitro, indicating an increase in the matrix composition similar to normal IVD [56]. In the study by utilizing human juvenile chondrocytes in a rat tail disc degeneration model, the disc injected with juvenile chondrocytes maintained the disc morphology and increased MRI index at 12 weeks post injection [57]. In ex-vivo degenerative disc disease (DDD) mimicking conditions, injected NC fully localize in NP without any visible signs of leakage and potential to stabilize NP structure with long term efficacy [58]. The autologous condrocytes were considered as valuable clinical tools in damaged IVD as they upregulated the cell viability, matrix production, and integration [59]. Further, there was a phase I clinical trial initiated employing juvenile articular chondrocytes delivered in fibrin carrier that reported preliminary outcomes in 15 patients in 2013 [60]. Likewise, in a clinical pilot study for disc repair by utilizing autologous chondrocytes, the patients demonstrated matrix regeneration and pain relief after cell transplantation [61]. Hence, these approaches indicate the regenerative potential of cartilage-derived cells in IVD regeneration. 

The self-assembly and remodeling process is a fundamental aspect of the mode of action of injectable engineered cartilage gel. This step generally assumes that cartilage gel goes through continuous matrix secretion and construction of collagen cross-links, yet this has seldom been demonstrated. The N-cadherin activity is considered to act a vital role in the initial cellular aggregation and self-assembly period during the in vitro engineering period [62]. Previous in vitro studies using chondrocytes and MSCs have found the upregulation of GAG production after day 14 and rise in collagen after day 28, while cartilage gel expressed both GAG and collagen at much earlier time points [62]. The mechanism behind the remodeling process after transplantation is not well known. A previous study using synovium derived MSCs indicated partial chondrogenic differentiation of the scaffold-free cartilage construct after transplantation, even without predifferentiation. It is found that, after transplantation of the cartilage gel, continued differentiation has been observed excessively due to commitment of hFCPCs towards chondrogenesis [28]. The assumption underlying the efficacy of cartilage gel is the continued physical presence of the transplanted construct within the defect, together with ongoing remodeling towards cartilage tissue after transplantation. The cohesive strength associated with the degree of the cartilage gel’s viscoelasticity may have played a role in the stickiness of our constructs to the defect area [63]. However, until now, the effect of cartilage gel after intra-articular transplantation was largely unknown, both in clinical and preclinical studies [28].

In our study, we focused only on NP preservation by the intradiscal injection of cartilage gel, but not on the AF. Although this provides temporary symptomatic relief, the AF defect remains untreated, which may cause reherniation and further degeneration of the IVD. In previous studies, individual NP and AF repair therapies have been explored as means to prevent IVD degeneration and promote healing after discectomy. Both therapies effectively treated their respective target tissue; however, the complex nature of the IVD makes a single treatment approach for IVD degeneration insufficient. In a study that combined a collagen AF patch with an HA NP injection, individual repairs targeting the NP or AF successfully treated their respective tissue, where the HA NP injection restored hydration to the NP, and the collagen AF patch prevented herniation through new tissue formation at the defect site. However, because neither of the individual repairs addressed both the hydration of the NP and the integrity of the AF, only treatment with the combined therapy resulted in IVDs that resembled healthy discs. The combined therapy filled defects in the AF, restored water content to the NP, maintained disc height and the native IVD morphology, and yielded functional mechanical properties similar to those of healthy discs [16]. Thus, in the future, the combined NP approach with injectable cartilage gel and the AF approach may be a potential platform that can maintain IVD health and prevent subsequent progressive degeneration in the spine. We performed the nucleotomy model in Co 4-5/5-6 in rat tail discs to demonstrate the regenerative characteristics of the cartilage gel by just incising the AF, without making an AF defect model. We demonstrated the regenerative efficacy of the cartilage gel without the use of scaffold. So far, scaffolds have been used to prevent cell leakage. However, scaffolds for disc repair are clinically not available because the repair mechanisms have yet to be elucidated and safety concerns persist. An inflammatory reaction may often be induced after implantation of scaffolds [17]. The scaffolds incorporated with tissue-engineered cartilage often make the construct relatively rigid and fixed in geometry, frequently forming a gap. Further, scaffolds may be weak, not enough to withstand the pressing force of IVDs. They may further destroy the healthy host tissue, and overall, they maximize the complexity in the surgical procedure [64]. Therefore, the injectable cartilage gel analyzed in this study would be another option for treating IVD degeneration in the future.

This study has several limitations. The rat tail discs are biologically and mechanically contra-distinct from rat lumbar discs, as well as humans [13]. For the generation of scaffold-free tissue-engineered cartilage, a larger number of cells is demanded in comparison to scaffold-based tissue-engineered cartilage [65,66]. The notochordal cells are retained in the disc NP of rodents throughout their lives, resulting in fewer age-related disc pathologies [67]. In addition, the surgical induction of nucleotomy-induced disc disruption in rat tails does not entirely mimic the clinical situation of the post-discectomy state in humans [68]. Therefore, preclinical studies using larger animal models in which disc NP notochordal cells disappear should be imposed with cartilage gel. Furthermore, we did not carry out biomechanical tests to investigate the dynamic viscoelastic properties of IVDs [69]. Similarly, a higher possibility of reherniation and further degeneration of discs is considerable, as we targeted repair of the NP but not the partially damaged AF. Similarly, the short experimental period of this study is insufficient and future studies should extend the experiment duration to a longer time to reflect the complete effects of the cartilage gel transplantation on the repair of degenerative IVD. Lastly, an important consideration is the availability of hFCPCs and ethical issues concerning the clinical use of fetal cells [70]. Although the fetal source of hFCPCs is sparse, they are obtained in large amount from a single donor (approximately 6.5 × 10^7^ cells/g tissue). hFCPCs proliferate well enough to establish cell banks for commercial use, thereby we have the prevalence of technology using cells from a small number of fetuses to many patients [29]. Likewise, we are legally using stillborn fetuses for various reasons. There are ethical concerns regarding the fetal source of hFCPCs but it is legally allowed for its sourcing in many countries.

To the best of our knowledge, no other study has explored the effect of a cartilage gel on IVD repair. Additionally, to figure-out the workability of applying cartilage gel to humans, it is mandatory to test the regenerative effects of cartilage gel by using larger animal models. Despite some limitations, this investigation demonstrates that regeneration following discectomy can be achieved by implantation of a cartilage gel.

## 4. Materials and Methods

### 4.1. Cell Isolation and Culture

All protocols that involved human tissue were carried out under the approval of the Institutional Review Board of Ajou University School of Medicine (AJIRB-MED-SMP-10-268). Human fetal cartilage tissues were obtained after informed consent from guardians from a single abortus. Using a previously published protocol, cells were isolated from the femoral head cartilage and expanded [29]. Cartilage tissue was minced into tiny pieces and treated with 0.1% collagenase type 2 (Worthington Biochemical Corp., Freehold, NJ, USA), in high-glucose Dulbecco’s Modified Eagle Medium (DMEM) containing 1% bovine serum albumin (BSA) at 37 °C under 5% CO_2_. After 12 h, the liberated cells were centrifuged at 1700 rpm for 10 min, washed twice, and cultured in DMEM supplemented with 10% FBS, 100 U/mL penicillin G, and 100 µg/mL streptomycin at a density of 8 × 10^3^ cells/cm^2^. Then, 0.5% trypsin-EDTA (10× to 1×; Gibco, NY, USA) was employed to detach the cells when they reached 80% confluence, after which they were replaced in the same way as above. Cells were expanded for two passages with the culture medium changed every 3 days. As previously published, all donor cells demonstrated no significant variation in terms of morphology, proliferation, surface marker, senescence, and chondrogenesis [29]. Passage 7 cells were used for all groups.

### 4.2. Fabrication of Cartilage Gel

The injectable engineered cartilage gel was manufactured using a previously published protocol [71]. In summary, obtained hFCPCs were cultured in a high-density monolayer at 2 × 10^5^ cells/cm^2^ using differentiation medium containing DMEM supplemented with 100 nM dexamethasone, 50 µg/mL ascorbate-2 phosphate, ITS supplement, 40 µg/mL proline, 1.25 mg/mL BSA, 100 µg/mL sodium pyruvate, and 10 ng/mL TGF-β1. Cells were cultured two-dimensionally until full confluency, when they spontaneously generated a thin membrane. The medium was removed and subsequently, 1× trypsin-EDTA was added to the thin membrane and incubated for less than 5 min at 37 °C. The enzymes were immediately removed when the membrane was peeled off from the plate and, with attention and care, the membrane was retrieved using a wide-bore pipette and transferred individually to a 50 mL tube filled with 10 mL serum-free DMEM. After that, centrifugation was done at 100× *g* for 20 min to each tube to consolidate the membrane into a pellet-type construct. Incubation was carried out for the constructs for 16 h at 37 °C in the differentiation medium above and after that transferred to a 6-well plate for extended culture for 2 weeks in a 37 °C humidified atmosphere of 95% air and 5% CO_2_. Every 3 days, the culture medium was transferred.

### 4.3. Animals 

Thirty female Sprague-Dawley rats (220–240 g), at 8 weeks of age, were purchased from Orient Bio. Inc. (Seongnam, Korea), and were acclimatized for a week at a light/dark cycle of 12/12 h (temperature: 22 ± 1°C and relative humidity; 50% ±1%) and free approach to food and water. The animal experiments were performed according to the direction approved by the Institutional Animal Care and Use Committee (IACUC) of CHA Bundang Medical Center (IACUC 200141). 

Prior to surgery, the peritoneal site was sterilized with 70% alcohol, and rats were deeply anesthetized with a general anesthesia mixture of Zoletil^®^ (50 mg/kg, Virbac Laboratories, Carros, France) and Rompun^®^ (10 mg/kg, Bayer, Korea) injected intraperitoneally. Then, the proximal-most part of the tail, along with the pelvic area, was sterilized with 70% alcohol followed by a povidone-iodine solution. A longitudinal incision of 1 cm was made along the tail to expose the lateral portion of the coccygeal disc. Subsequently, a #11 scalpel blade was inserted 1.5 mm into the coccygeal disc (Co4-5, Co5-6), and then, nucleotomy was performed by disc AF incision and NP aspiration with a 22-gauge catheter on a 5-mL syringe [13]. Nucleotomy at Co4-5 was performed to assess the effects of materials in 30 rats. Thereafter, 15-μL intradiscal injections of cartilage gels, hFCPCs, and ECM were injected using a 25-gauge catheter (Figure 1A). ECM was prepared by freeze-thawing the cartilage gel five cycles of liquid nitrogen and 37 °C water bath for 1 min each thereby consisting only with cartilage matrix and dead cells of the cartilage gel. We did not inject anything after nucleotomy in Co5-6 (the injury-only group). Finally, the skin was sutured and disinfected, and an appropriate dose of analgesic (Ketoprofen, SCD Pharm. Co. Ltd., Seoul, Korea) and antibiotic (Cefazolin, CKD Pharmaceuticals, Seoul, Korea) for 3 days after surgery was provided. Throughout the surgical procedure, the rats’ body temperature was maintained at 37 °C with the aid of a thermostatically regulated heating pad. After 6 weeks of implantation, the rats were euthanized by CO_2_ asphyxiation and coccygeal discs were removed for radiologic, histologic, and mRNA analyses.

### 4.4. Quantitative Behavioral Nociception Assays

The von Frey test was carried out 2 days before surgery (day-2) and 2, 7, 14, 21, 28, 35, and 42 days after the surgery. The rats were individually placed into a six-compartment rat enclosure with wire mesh floors and lids with air holes for a 20-min habituation period to avoid exploratory activities of rats. After that, on the ventral surface of the tail, a 2-g filament was applied for a maximum of 6 s with sufficient force. Responses such as flinching, licking, withdrawing, or shaking the base of the tail immediately or within 6 s were considered positive responses. However, if the animals did not show any responses when the filament was applied, then it was considered a negative response. This process was repeated five times for the 2-g filament. If a response was obtained with the 2-g filament, then the animal was tested with progressively lower-weight filaments until no response was obtained in five attempts. The test was continued with ascending filament numbers until five responses were obtained in five attempts for two consecutive filaments. The experiment began by testing the response to a filament estimated to be close to the 50% withdrawal threshold [48]. Two independent observers who were blinded to the specimen’s treatment were involved in the von Frey analysis.

### 4.5. Magnetic Resonance Imaging 

Six weeks after implantation in the rat coccygeal disc, we performed a 9.4 T magnetic resonance imaging (MRI) examination (Bruker BioSpec, Billerica, MA, USA) to study the changes in the structure of the disc, the extent of degeneration of the coccygeal disc, and existence of water content in the disc. T2-weighted imaging for (1) the coronal plane was performed as follows: time to repetition (TR) of 5000 ms, time to echo (TE) of 30 ms, 150 horizontal_150 vertical matrix; field of view of 15 horizontal × 15 vertical, and 0.5-mm slices with 0-mm spacing between each slice; (2) the sagittal plane: TR of 5000 ms, TE of 30 ms, 150 horizontals_150 vertical matrix; field of view of 15 horizontal × 15 vertical, and 0.5-mm slices with 0-mm spacing between each slice. The MRI index (calculated as the area of NP multiplied by average signal intensity) was calculated to evaluate the degree of degeneration of the coccygeal discs [72]. The high signal intensity area in the mid-coronal plane of the T2 weighted images was considered as the region of interest (ROI), as the outline of the NP. The ROI was measured using ImageJ software (Version 1.50b, National Institutes of Health, Bethesda, MD, USA). The Pfirrmann classification, with grades ranging from I (normal) to V (advanced degeneration), was used to access the degree of disc degeneration [73]. To measure the MRI index and Pfirrmann grades, the two independent observers who were blinded by the specimen’s treatment were involved.

### 4.6. Histological Analysis 

Six weeks after implantation, rats were sacrificed, and discs were harvested for histological analysis to demonstrate proteoglycan distribution in the IVD. The disc with the adjacent vertebral body was fixed in 10% neutral buffered formalin for 1 week and decalcified in Rapid Cal Immuno (BBC Biochemical, Mount Vernon, WA, USA) for 2 weeks. Tissues were then processed for paraffin embedding and sectioning into coronal sections (10 µm) using a microtome (Leica, Wetzlar, Germany). The obtained sections were dewaxed, rehydrated, and stained with safranin-O (Sigma, St. Louis, MO, USA) to analyze the quantity and distribution of proteoglycan content. Finally, the sections were mounted using mounting media and scanned with an Olympus C-mount camera adapter (U-TVO.63XC, Tokyo, Japan). 

Likewise, the histological scoring was done by using a comprehensive 16-point scale for the assessment of IVD based on safranin-O staining. The scoring was based on the NP morphology, NP cellularity, AF morphology, endplate morphology, and the boundary between the NP and AF, resulting in five subcategories. Briefly, the NP and AF morphology each include two degenerative features since they were ranked to be highly crucial in the study. As alterations in notochordal cell morphology are an important and easily notable feature in degenerative rat IVDs, so the NP cellularity category was also weighted twice with two features. Briefly, in terms of NP morphology, analyses on the NP shape and total NP area were performed. Similarly, cell number and cellular morphology were analyzed to measure NP cellularity. The border appearance was observed in order to differentiate no interruption, minimal interruption, and no distinction between NP and AF. Furthermore, to examine AF morphology in IVD, lamellar organization and any tears/fissures/disruptions in the AF region were identified. Finally, in terms of endplate, any appearance of disruptions, microfractures, osteophyte, or ossifications were examined to produce the histological scores. Thus, non-degenerative characteristics were represented as 0, mild degenerative characteristics as 1, and severe degenerative changes as 2. The sum of the separate scores ranged from 0 (normal) to 16 (most severe) The NP, AF, endplate, and boundary between the NP and AF of the IVD were scored and added together for a total IVD score [74]. Two independent observers who were completely blinded to the sample information conducted the histological analysis of all samples. 

Furthermore, the obtained sections were dewaxed, rehydrated, and stained with H&E for analysis of the tissue morphology and proteoglycan distribution in IVDs. The disc NP-cell number and H&E positive area were measured using ImageJ software (https://imagej.nih.gov/ij/ (accessed on 10 November 2022)). Briefly, we created binary images at a fixed intensity level and measured the area between vertebral endplates [13]. 

### 4.7. Immunofluorescence and Immunohistochemistry

Rats were euthanized via excess carbon dioxide inhalation, and the coccygeal discs were collected 6 weeks after implantation, and an immunohistochemical analysis was performed for aggrecan and type II collagen. We performed immunofluorescence analysis for a calcitonin gene receptor protein (CGRP), Tie2, brachyury, matrix metalloproteinase-13 (MMP-13), human nuclei antibody, tumor necrosis factor-alpha (TNF-α), and interleukin-1-beta (IL-1β). The harvested tissues were fixed overnight in a 4% paraformaldehyde (PFA) solution and decalcified by using a decalcification solution (RapidCal Immuno; BBC Biochemical, Mount Vernon, WA, USA) for 2 weeks. Then, discs were embedded within paraffin wax and sectioned longitudinally using a microtome (Leica) into sections of 5–10 µm thickness. For the immuno-staining, the first sections were dewaxed, rehydrated, and after that stained with primary antibodies against aggrecan (ab36861; Abcam, Cambridge, UK, 1:1000), type II collagen (ab34712; Abcam, Cambridge, UK, 1:100), CGRP (ab47027; Abcam, Cambridge, UK, 1:200), Tie2 (NBP2-20636; Novus Biologicals, Littleton, CO, USA, 1:200), brachyury (sc-166962; Santa Cruz, Dallas, TX, USA, 1:200), human nuclei antibody (MAB1281; Sigma Aldrich, St. Louis, MO, USA, 1:200), TNF-α (ab6671; Abcam, Cambridge, UK, 1:200), MMP-13 (ab39012; Abcam, Cambridge, UK, 1:200), and IL-1β (AF-501-NA; Novus Biologicals, Littleton, CO, USA, 1:200). Then, after 24 h of incubation, sections were washed with phosphate-buffered saline with Tween 20 and again incubated with the secondary antibody anti-Rb horseradish peroxidase (Roche Diagnostics Ltd., Basel, Switzerland), and Alexa Fluor 488 (A11034; Invitrogen, Waltham, MA, USA, 1:400), Alexa Fluor 488 (A11029; Invitrogen, Waltham, MA, USA, 1:400), Alexa Fluor 568 (A10042; Invitrogen, Waltham, MA, USA, 1:400), and Alexa Fluor 647 (A21469; Invitrogen, Waltham, MA, USA, 1:400)-conjugated secondary antibodies. After that, specimens were carried out for the washing step, then the specimens were counterstained with 4′, 6-diamidino-2-phenylindole (DAPI) (Abcam, Cambridge, UK, 1:1000) and incubated for 10 min. The sections were mounted and finally examined using a fluorescence microscope (Zeiss 880, Oberkochen, Germany, and Leica SP5, Wetzlar, Germany). The percentages of the positive area for aggrecan and type II collagen and positive cell number relative to DAPI for CGRP, Tie 2, brachyury, MMP-13, human nuclei antibody, TNF-α, and IL-1β were calculated using ImageJ software (Version 1.50b, https://imagej.nih.gov/ij/ (accessed on 10 November 2022)).

### 4.8. RNA Isolation and Real-Time RT-PCR

The coccygeal discs were harvested six weeks after implantation, and the NP was isolated from the disc and triturated under liquid nitrogen in a pre-cooled mortar. The liquid nitrogen was allowed to volatilize and, with the assistance of a pestle, the hardened NP was ground into a fine powder. At 37 °C for 10 min, the viscous NP was then incubated in 1 mL of Trizol (Invitrogen, Carlsbad, CA, USA). Then, 0.2 mL of chloroform was added and centrifuged at 12,000× *g* and 4 °C for 15 min. Following centrifugation, the upper layer was transferred to 0.5 mL isopropanol for precipitation, and centrifugation was performed again. After that, the supernatant was removed, and the RNA precipitate was washed once with 75% alcohol and dried at 37 °C for 10 min. Then, the obtained RNA was dissolved by using 20 µL of RNAase-free water. To perform polymerase chain reaction (PCR) analysis, a Maxime™ RT premix Kit (iNtRON Biotechnology, Seoul, Korea) was used. First, 1 µg of RNA was reverse-transcribed to complementary DNA. After that we performed real-time PCR by using SYBR Green Master Mix, and the ABI Step One Real-Time PCR system (Applied Biosystems, Waltham, MA, USA) was used to analyze the mRNA expression. The typical PCR amplification profile used was denaturation at 95 °C for 10 min followed by a second step at 95 °C for 15 s, followed by annealing and extension at 60 °C for 30 s, and a melting curve analysis at 40 cycles, in which dissociation curve software was used to ensure that only a single product was amplified. Target genes were normalized with glyceraldehyde 3-phosphate dehydrogenase (GAPDH), and the data were analyzed by the 2^-ΔΔct^ method. Primer sequences for the gene of interest used in this study are provided in Table 1. 

### 4.9. Statistical Analysis

For the statistical analysis of the data, GraphPad Prism (version 5.01, GraphPad Software) was used, and ImageJ software (Version 1.50b, https://imagej.nih.gov/ij/ (accessed on 10 November 2022)) was used for the quantification of data. Data are presented as mean, ±standard error of the mean (SEM), box plot, and One-way analysis of variance (ANOVA). Bartlett’s test was used to assess the effects of multiple treatments in in vivo experiments, and *p*-values < 0.05 were considered statistically significant.

## 5. Conclusions

In conclusion, this study addressed the efficacy of an injectable engineered cartilage gel for IVD regeneration in comparison with its source cells (hFCPCs) or ECM components alone. The cartilage gel enhanced IVD regeneration by increasing cell density, reducing radiological and histological damage, upregulating ECM proteins such as aggrecan and type II collagen, and downregulating pro-inflammatory cytokines, such as TNF-α and IL-1β. The therapeutic effect of the cartilage gel on IVD regeneration was much superior to that of its individual components (hFCPCs and cartilage ECM). This study demonstrated the high potential of cartilage gel implantation as an innovative treatment for IVD herniation, as it not only prevented IVD tissue degeneration but also reduced pain. Furthermore, in the future, simple intradiscal injections of injectable engineered cartilage gel may be an effective treatment option without any scaffolds or growth factors for IVD diseases.

## Figures and Tables

**Figure 1 ijms-24-03146-f001:**
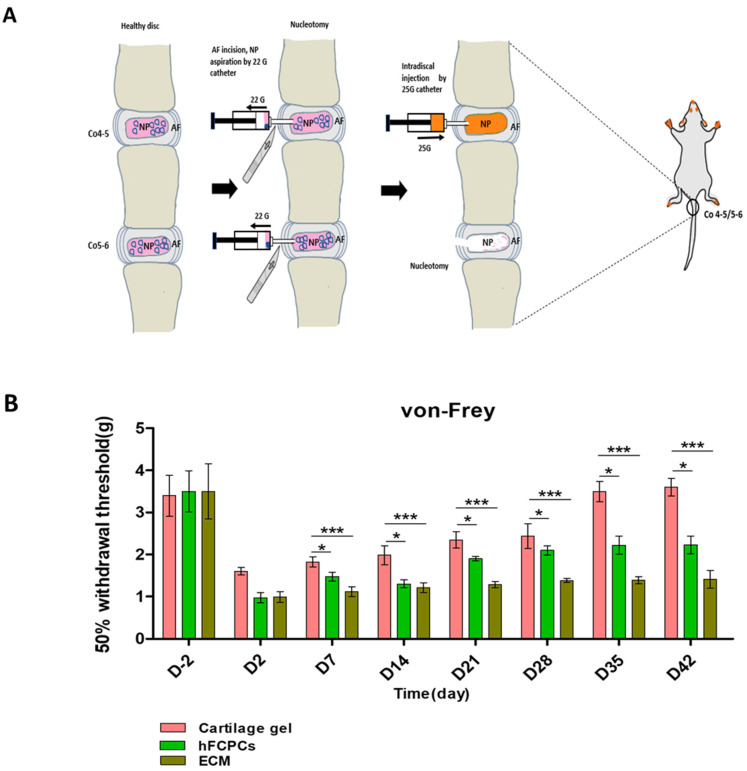
Schematic diagram of the surgical procedure and alleviation of the injury-induced pain phenotype by the cartilage gel. (**A**) Surgical procedures in a rat tail nucleotomy model; nucleus pulposus aspiration with a 22-gauge catheter through an annulus fibrosus incision. Injection of cartilage gel, hFCPCs, or ECM (15 µL using a 25-gauge catheter). (**B**) Alleviation of the injury-induced pain phenotype by the cartilage gel. In the von Frey test, the 50% withdrawal thresholds were significantly higher in the cartilage gel-treated group than in the hFCPCs or ECM-treated group. * *p* < 0.05; *** *p* < 0.001, significant difference between groups, by one-way ANOVA, *n* = 10 rats per group. hFCPCs, human fetal cartilage-derived progenitor cells; ECM, extracellular matrix.

**Figure 2 ijms-24-03146-f002:**
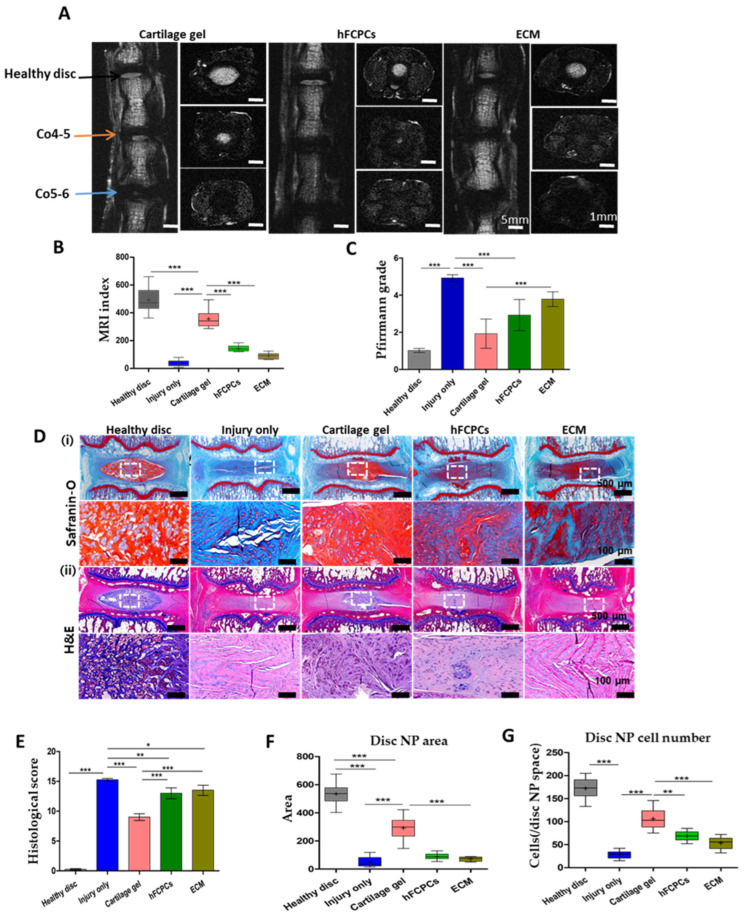
Cartilage gel injection promoted the repair of degenerated discs by restoring disc hydration and proteoglycan content in a rat tail nucleotomy model. The Co3-4 discs were healthy controls, the Co5-6 discs were the nucleotomy-only group, and the Co4-5 discs were the groups injected with the cartilage gel, hFCPCs, or ECM after nucleotomy. (**A**) T2-weighted magnetic resonance imaging (MRI) of rat tail discs of experimental groups taken at 6 weeks after nucleotomy. Black arrows indicate healthy discs (control), orange arrows indicate injected discs (Co4-5), and blue arrows indicate nucleotomy-only discs (Co5-6). (**B**) Changes in the T2-weighted MRI index. (**C**) Changes in the Pfirrmann grade as assessed by T2-weighted MRI. (**D**) (i) Safranin-O (S-O) staining of rat tail discs at 6 weeks after nucleotomy in low-power fields (top). White rectangles indicate the disc nucleus pulposus (NP) area shown in higher-power fields (bottom). (ii) Changes in hematoxylin and eosin (H&E) staining of disc NP spaces (top). White rectangles indicate the disc NP area shown in higher-power fields (bottom). (**E**) Changes in the histological score by S-O staining. (**F**) Changes in the positive area of H&E staining in disc NP spaces. (**G**) Changes in the number of NP cells in the disc NP spaces measured using H&E images. In (**B**,**F**,**G**), data are presented with box plots (*n* = 7, *n* = 6 and *n* = 6 respectively). In (**A**,**E**), data are presented with mean ± SEM (*n* = 7 and *n* = 6, respectively, * *p* < 0.05; ** *p* < 0.01; *** *p* < 0.001). significant difference by one-way ANOVA. hFCPCs, human fetal cartilage-derived progenitor cells; ECM, extracellular matrix.

**Figure 3 ijms-24-03146-f003:**
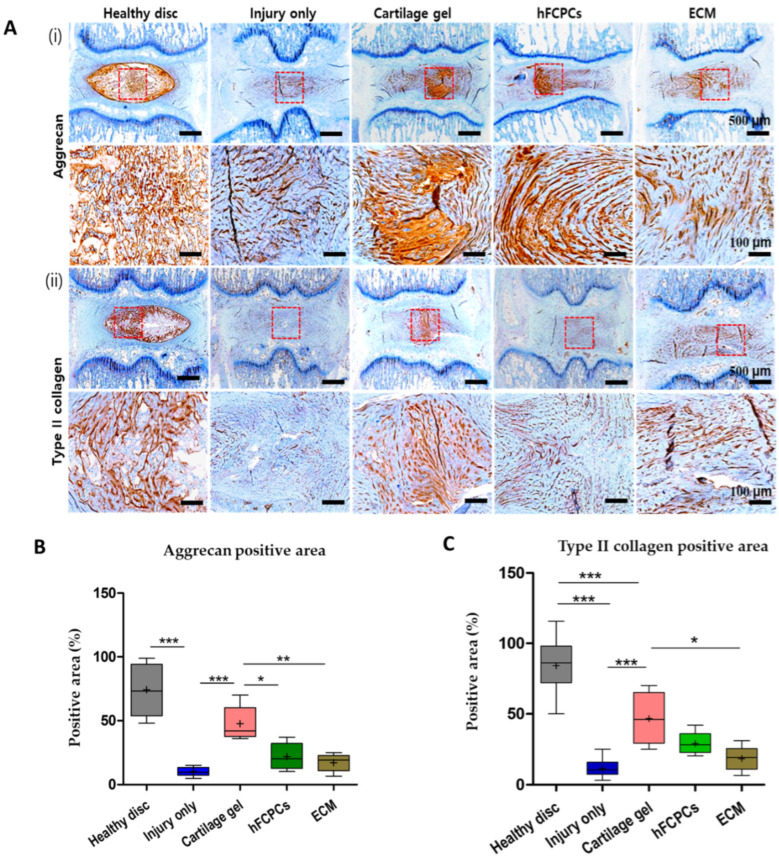
The cartilage gel preserved the matrix proteins in disc nucleus pulposus (NP) of a rat nucleotomy model. Immunohistochemical images of the indicated genes are presented for the healthy discs, injury-only discs, and the cartilage gel-injected, hFCPCs-injected, or ECM-injected discs at 6 weeks after surgery. (**A**) Immunohistochemistry of aggrecan (i) and type II collagen (ii). (**B**) Changes in the percentage of aggrecan-positive area in the disc NP spaces. The whole IVD images with merged signals are also presented at the bottom. (**C**) Changes in the percentage of type II collagen-positive area in the disc NP spaces. In (**B**,**C**), data are presented with box plots (*n* = 6, * *p* < 0.05; ** *p* < 0.01, *** *p* < 0.001, significant difference by one-way ANOVA). hFCPCs, human fetal cartilage-derived progenitor cells; ECM, extracellular matrix.

**Figure 4 ijms-24-03146-f004:**
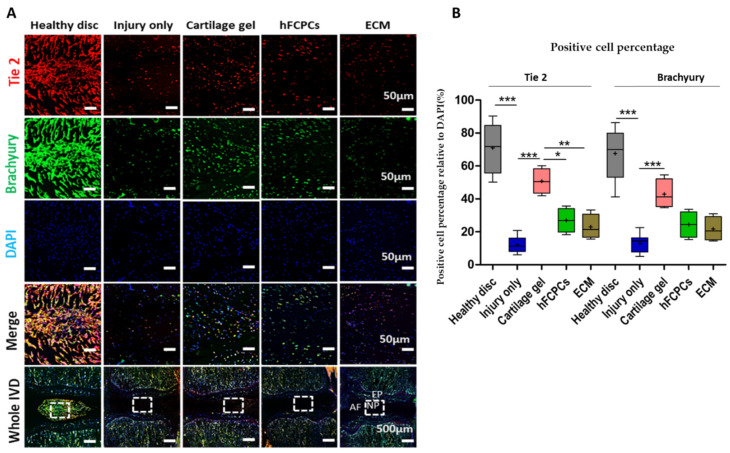
The cartilage gel preserved cells with endogenous phenotype in disc nucleus pulposus (NP) of a rat nucleotomy model. Immunofluorescence images of the indicated genes are presented for the healthy discs, injury-only discs, and the cartilage gel-injected, hFCPCs-injected, or ECM-injected discs at 6 weeks after surgery. (**A**) Immunofluorescence of brachyury (green) and Tie2 (red), DAPI (blue), and merged signals. (**B**) Changes in the percentage of Tie2 or brachyury-positive cells in the disc NP spaces. Immunopositivity was counted in disc NP of low-power fields and calculated as relative to the total number of DAPI-positive cells. In (**A**,**B**), data are presented with box plots (*n* = 4, * *p* < 0.05; ** *p* < 0.01, *** *p* < 0.001, significant difference by one-way ANOVA). hFCPCs, human fetal cartilage-derived progenitor cells; ECM, extracellular matrix.

**Figure 5 ijms-24-03146-f005:**
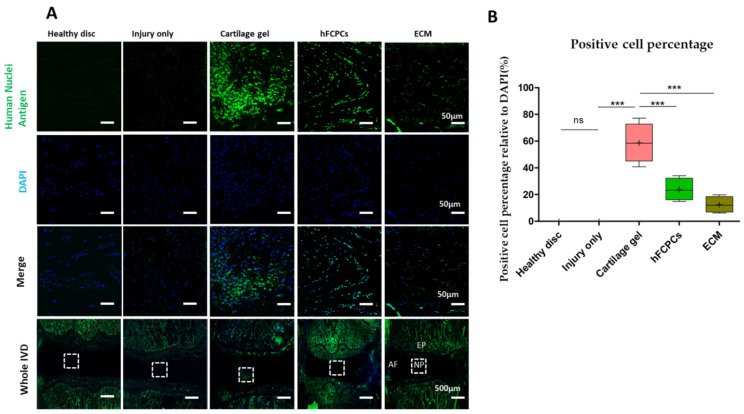
Preservation of implanted cells in the disc nucleus pulposus (NP) of a rat tail nucleotomy model. (**A**) Immunofluorescence of rat tails in the cartilage gel-injected, hFCPCs-injected, ECM-injected, healthy, and injury-only discs at six weeks after nucleotomy for human nuclei antigen (green), DAPI (blue), and merged signals. (**B**) Changes in the percentage of human nuclear antigen-positive cells. Immunopositivity was counted in the disc NP with low-power fields and calculated as relative to the total number of DAPI-positive cells. Data are presented with box plots (*n* = 4, *** *p* < 0.001, ns, not significant, significant difference by one-way ANOVA). hFCPCs, human fetal cartilage-derived progenitor cells; ECM, extracellular matrix.

**Figure 6 ijms-24-03146-f006:**
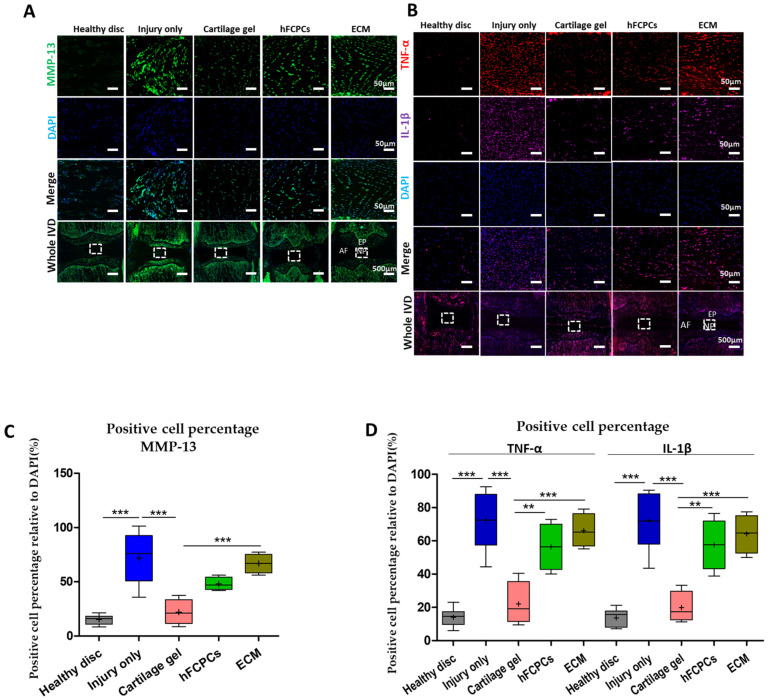
Cartilage gel downregulated catabolic and proinflammatory cytokines in the disc nucleus pulposus (NP) of a rat tail nucleotomy model. Immunofluorescence images of the indicated genes are presented for the healthy discs, injury-only discs, and the cartilage gel-injected, hFCPCs-injected, or ECM-injected discs at six weeks after surgery. (**A**) Immunofluorescence of the rat tail discs for matrix metalloproteinase-13 (MMP-13) (green), DAPI (blue), and merged signals. (**B**) Immunofluorescence of rat tail discs for tumor necrosis factor-alpha (TNF-α) (red) and interleukin (IL)-1β (purple), DAPI (blue) and merged signals. (**C**) Changes in the percentage of MMP-13-positive cells in the disc NP spaces. (**D**) Changes in the percentage of TNF-α- or IL-1 β-positive cells in the disc NP spaces. In (**C**,**D**) immunopositivity was counted in the disc NP with low-power fields and calculated as relative to the total number of DAPI-positive cells. Data are presented with box plots (*n* = 4, ** *p* < 0.01, *** *p* < 0.001, significant difference by one-way ANOVA).

**Figure 7 ijms-24-03146-f007:**
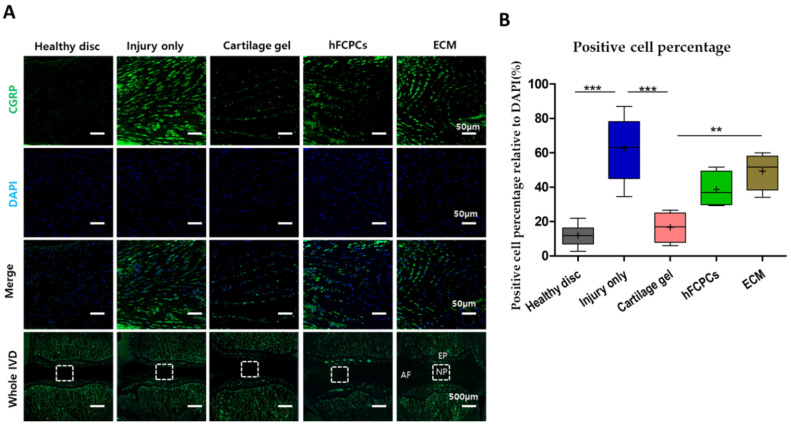
Cartilage gel inhibited injury-induced pain in the disc nucleus pulposus (NP) of a rat tail nucleotomy model. (**A**) Immunofluorescence of rat tails of cartilage gel-injected, hFCPCs-injected, ECM-injected, healthy, and injury-only discs at six weeks after nucleotomy for CGRP (green), DAPI (blue), and merged signals. (**B**) Changes in the percentage of CGRP-positive cells in the disc NP spaces. Immunopositivity was counted in the disc NP with low-power fields and calculated as relative to the total number of DAPI-positive cells. Data are presented with box plots (*n* = 4, ** *p* < 0.01, *** *p* < 0.001, significant difference by one-way ANOVA). hFCPCs, human fetal cartilage-derived progenitor cells; ECM, extracellular matrix; CCRP, calcitonin gene receptor protein.

**Figure 8 ijms-24-03146-f008:**
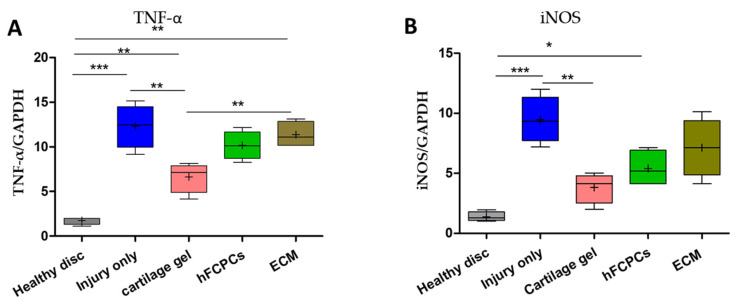
Cartilage gel downregulated mRNA levels of proinflammatory cytokines in the disc nucleus pulposus (NP) of a rat tail nucleotomy model. (**A**) mRNA levels of TNF-α in the disc NP spaces. (**B**) mRNA levels of iNOS in the disc NP spaces. In (**A**,**B**), data are presented in box plots (*n* = 3, * *p* < 0.05; ** *p* < 0.01; *** *p* < 0.001, significant difference by one-way ANOVA).

**Table 1 ijms-24-03146-t001:** Sequences of primers for real-time RT-PCR analysis.

Primer	Directions	Sequences
GAPDH	Forward	5′-CAACTCCCTCAAGATTGTCAGCCAA-3′
	Reverse	5′-GGCATGGACTGTGGTCATGA-3′
iNOS	Forward	5′-CTGCAGGTCTTTGACGCTCGAG-3′
	Reverse	5-GTGGAACACAGGGGTGATGATCTCC-3′
TNF-α	Forward	5′-AAATGGGCTCCCTCTATCAGTTC-3′
	Reverse	5′TCTGCTTGGTGGTTTGCTACGAC-3′

## Data Availability

All datasets used and/or analyzed during the current study are available from the corresponding author on reasonable request.

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
