# Peer review of "An Injectable Engineered Cartilage Gel Improves Intervertebral Disc Repair in a Rat Nucleotomy Model"

_ijms, 2023, doi:10.3390/ijms24043146_

Round 1

Reviewer 1 Report

This is a very interesting study concerning the regeneration potential of injectable cartilage gel to rescue the degenerative disc. The manuscript is well written.

However, the appearance of figures should follow time sequence in the text.

For example, Figure 3B and E appeared after Figure 4. Figure 4E and D followed after Figure 6. They should be revised.

Author Response

Reviewer 1 comments:

#1 This is a very interesting study concerning the regeneration potential of injectable cartilage gel to rescue the degenerative disc. The manuscript is well written.

However, the appearance of figures should follow time sequence in the text.

For example, Figure 3B and E appeared after Figure 4. Figure 4E and D followed after Figure 6. They should be revised.

Response #1:

Thank you very much for the reviewer’s suggestions and great efforts toward improving our manuscript. We have revised the appearances of all the figures in the manuscript.

Reviewer 2 Report

In this article, researchers investigated the effectiveness of an engineered cartilage gel utilizing human fetal cartilage-derived progenitor cells (hFCPCs) on intervertebral disc repair in a rat tail nucleotomy model. Their results demonstrate that cartilage gel has perfect therapeutic potential for intervertebral disc injury. I believe readers in this field will gain a lot from this paper. However, before acceptance some minor issues should be carefully addressed.

1. The rheological properties of the cartilage gel should be measured.

2. The authors should show us multi-directional differentiation potential of human fetal cartilage-derived progenitor cells.

Author Response

Response to the reviewer

[Reviewer 2]

In this article, researchers investigated the effectiveness of an engineered cartilage gel utilizing human fetal cartilage-derived progenitor cells (hFCPCs) on intervertebral disc repair in a rat tail nucleotomy model. Their results demonstrate that cartilage gel has perfect therapeutic potential for intervertebral disc injury. I believe readers in this field will gain a lot from this paper. However, before acceptance some minor issues should be carefully addressed.

#1. The rheological properties of the cartilage gel should be measured.

Response #1

We are greatly thankful to the reviewer’s comments.

We have previously published the information about the aggregate modulus, spreadability and adhesion strength of the cartilage gel [28]. As shown in the figure below, the mechanical properties are changing along with the culture times of 1, 2 and 3 weeks in vitro. We used the samples cultured for 2 weeks in this study and provided the approximate values at 2 weeks in Introduction (page 3, lines 97-98).

  1. The authors should show us multi-directional differentiation potential of human fetal cartilage-derived progenitor cells.

Response #2

Thank you so much for this comment. hFCPCs have a differentiation potential to 3 mesengenic lineages of chondrocytes, osteoblasts and adipocytes. Their differentiation ability was superior to adult chondrocytes and MSCs as shown below. We have published the data in 2 previous papers [28, 29]. We have a preliminary data showing hFCPCs do not differentiate to ectodermal and endodermal lineages.

Reviewer 3 Report

The present manuscript reports the used of HFCP cells derived gel for the regeneration of NP in an IVD defect model. The data is well presented and the manuscript is well written over all. I have the following few suggestions before the manuscript could be accepted:

- Methods

-- Please add some detail about the histological scoring method

-- What exactly do you mean by ECM (the third treatment group)? Is it decellularized cartilage? If so how was it prepared?

-- I see that the replicates (n) range from 3-10 in different experiments. Did the authors run any normality tests before using the parametric statistical test (ANOVA)?

Results:

-- Did the authors try treating the IVD with HFCP cell seeded ECM to see if it is the gel that contributes to healing or just that cells require a scaffold to perform better?

-- Please add the MMP13 label in figure 4C as in D, E, and F

-- Move section 2.6 and 2.9 where rest of data in Figure 3 and 4 respectively is discussed.

Discussion

-- Is it viable to use HFCP cells for future clinical use, in terms of abundance of source tissue and ethical concerns?

Author Response

Response to the reviewer

Reviewer 3 comments

The present manuscript reports the used of HFCP cells derived gel for the regeneration of NP in an IVD defect model. The data is well presented and the manuscript is well written over all. I have the following few suggestions before the manuscript could be accepted:

Methods

#1 Please add some detail about the histological scoring method

Response #1:

Thank you so much for this comment. 

The scoring was based on the NP morphology, NP cellularity, AF morphology, endplate morphology, and the boundary between the NP and AF resulting in five subcategories.

The NP and AF morphology each include two degenerative features since they were ranked to be highly crucial in the study. As alterations in notochordal cell morphology are an important and easily notable feature in degenerative rat IVDs, so the NP cellularity category was also weighted twice with two features. Briefly, in terms of NP morphology, analyses on the NP shape and total NP area were performed. Similarly, cell number and cellular morphology were analysed to measure NP cellularity. The border appearance was observed in order to differentiate no interruption, minimal interruption, and no distinction between NP and AF. Furthermore, to examine AF morphology in IVD, lamellar organization and any tears/fissures/disruptions in the AF region were identified. Finally, in terms of endplate, any appearances of disruptions, microfractures, osteophyte or ossifications were examined to produce histological scores. Thus, non-degenerative characteristics were represented as 0, mild degenerative characteristics were as 1, and severe degenerative changes were as 2. The sum of the separate scores ranged from 0 (normal) to 16 (most severe). The NP, AF, endplate, and boundary between the NP and AF of the IVD were scored and added together for a total IVD score [74].

 We added this information in the Materials and methods section 4.6 of our manuscript.  (Pages 20 and 21, lines 607-621)

#2 What exactly do you mean by ECM (the third treatment group)? Is it decellularized cartilage? If so how was it prepared?

Response#2:

We are thankful to the reviewer for this question.

The ECM were derived from the cartilage gel by freezing and thawing it

5 times cycles of liquid nitrogen and 37°C water bath. Therefore the ECM consists of cartilage-specific matrix and dead cells of cartilage gel. We added the information in Materials & Methods, section 4.3 (page 19, lines 553-554).

#3 I see that the replicates (n) range from 3-10 in different experiments. Did the authors run any normality tests before using the parametric statistical test (ANOVA)?

Response #3

Thank you so much for the reviewer’s comments. We had performed the Bartlett’s test. We have added in the materials and method, section 4.9 of our manuscript. (Page 22, line 683)

Results

#4 Did the authors try treating the IVD with HFCP cell seeded ECM to see if it is the gel that contributes to healing or just that cells require a scaffold to perform better?

Response #4:

We are very grateful for the reviewer’s comments.

We intended to see that hFCPCs or cartilage matrix (decellularized ECM) alone was not sufficient and their composite is important in healing the degenerative IVD. The cartilage gel is not a hFCPC-seeded ECM scaffold. It is a tissue engineered cartilage fabricated by 3D culture of hFCPCs and consists of hFCPCs and cartilage-specific ECMs secreted by hFCPCs themselves. Please refer to Materials & Methods, Section 4-2, and reference [28]. The adhesive nature of the cartilage gel on irregularly shaped defects would lead to a fixation-free, seamless fit into the defect area and high applicability (page 3, lines 98-100). Similarly, in an in vivo study in a nude mouse model and a non-human primate cartilage defect model, the scaffold-free cartilage gel showed promising results in repairing cartilage defect, cell distribution and remodeling process (page 3 lines 93-94). So far, scaffolds have been used mainly to prevent cell loss and mostly too weak to withstand the pressing force of IVDs. Therefore, the injectable cartilage gel used in this study would be another option for treating IVD degeneration in the future. Please refer to the descriptions in the manuscript (Page 17, lines 466-467,471-472,473-475).

#5 Please add the MMP13 label in figure 4C as in D, E, and F

Response#5:

We are very grateful for the reviewer’s suggestion. We have added the MMP-13 label in the figure.

#6 Move section 2.6 and 2.9 where rest of data in Figure 3 and 4 respectively is discussed.

Response #6:

We are thankful to the reviewer for the suggestion. We have made changes according to the reviewer’s comment.

Discussion

#7 Is it viable to use HFCP cells for future clinical use, in terms of abundance of source tissue and ethical concerns?

Response #7:

We are very grateful for the reviewer’s suggestion.

There are limitations in the source tissue but hFCPCs are abundant in the fetal cartilage yielding 20 folds more cells/g of tissue than adult cartilage tissue, therefore we can have the prevalence of technology using cells from small number of fetuses to many patients [29]. We could obtain large amount of hFCPCs to establish cell banks for commercial purpose and are currently developing therapeutics to treat knee cartilage defects and disc degeneration. There are ethical concerns regarding the fetal source of hFCPCs but as far as we understand there is no legal problem with its commercial use in many countries. Likewise, we are legally using stillborn fetuses for various reasons. These are added in Discussion (page 18, lines 491-496).